# Induced Relaxation Enhances the Cardiorespiratory Dynamics in COVID-19 Survivors

**DOI:** 10.3390/e25060874

**Published:** 2023-05-30

**Authors:** Alejandra Margarita Sánchez-Solís, Viridiana Peláez-Hernández, Laura Mercedes Santiago-Fuentes, Guadalupe Lizzbett Luna-Rodríguez, José Javier Reyes-Lagos, Arturo Orea-Tejeda

**Affiliations:** 1School of Medicine, Universidad Autónoma del Estado de México (UAEMéx), Toluca de Lerdo 50180, Mexico; asanchezs199@alumno.uaemex.mx (A.M.S.-S.); jjreyesl@uaemex.mx (J.J.R.-L.); 2Cardiology Service, Instituto Nacional de Enfermedades Respiratorias Ismael Cosío Villegas (INER), Mexico City 14080, Mexico; g.lizzbett.luna.rod@gmail.com (G.L.L.-R.); oreatart@gmail.com (A.O.-T.); 3Health Sciences Department, Universidad Autónoma Metropolitana Unidad Iztapalapa (UAM-I), Mexico City 09340, Mexico

**Keywords:** cardiorespiratory coupling, post-COVID-19 syndrome, slow breathing, diaphragmatic breathing, breathing and relaxation exercises, pulse–respiration quotient

## Abstract

Most COVID-19 survivors report experiencing at least one persistent symptom after recovery, including sympathovagal imbalance. Relaxation techniques based on slow-paced breathing have proven to be beneficial for cardiovascular and respiratory dynamics in healthy subjects and patients with various diseases. Therefore, the present study aimed to explore the cardiorespiratory dynamics by linear and nonlinear analysis of photoplethysmographic and respiratory time series on COVID-19 survivors under a psychophysiological assessment that includes slow-paced breathing. We analyzed photoplethysmographic and respiratory signals of 49 COVID-19 survivors to assess breathing rate variability (BRV), pulse rate variability (PRV), and pulse–respiration quotient (PRQ) during a psychophysiological assessment. Additionally, a comorbidity-based analysis was conducted to evaluate group changes. Our results indicate that all BRV indices significantly differed when performing slow-paced breathing. Nonlinear parameters of PRV were more appropriate for identifying changes in breathing patterns than linear indices. Furthermore, the mean and standard deviation of PRQ exhibited a significant increase while sample and fuzzy entropies decreased during diaphragmatic breathing. Thus, our findings suggest that slow-paced breathing may improve the cardiorespiratory dynamics of COVID-19 survivors in the short term by enhancing cardiorespiratory coupling via increased vagal activity.

## 1. Introduction

The COVID-19 disease, caused by the novel coronavirus SARS-CoV-2, emerged at the end of 2019 and has rapidly spread worldwide since it first appeared in China, resulting in more than 673 million cases and 6.8 million deaths at the time of writing this article [1,2,3]. The pathological response to the infection ranges from asymptomatic to severe cases, and the main complications caused by the disease include hypoxemic respiratory failure, heart failure, kidney injury, thromboembolic damage, and acute respiratory distress syndrome (ARDS), which represents most ICU admissions due to COVID-19 illness [4,5,6]. A significant proportion of individuals reported persistent symptoms beyond 12 weeks and up to 9 months that were not associated with any other diagnosis and occurred after the recovery from the acute phase of the disease [7,8], regardless of if the patient experienced an asymptomatic or severe case [9]. These manifestations are part of the post-acute sequelae of SARS-CoV-2 infection (PASC) or post-COVID syndrome, which involves respiratory, neurological, endocrine, cardiovascular, and other multisystemic alterations [10,11].

A high prevalence of autonomic alterations in respiratory and cardiovascular systems, which modify cardiopulmonary interactions as a sign of dysautonomia or orthostatic syndromes, occurs in subjects who suffered an acute phase of the disease [12,13,14]. Several signs and symptoms associated with this autonomic impairment have been reported, with chronic fatigue, chest pain, dyspnea, sleep disturbances, anxiety, depression, and post-traumatic stress syndrome being the most common among post-COVID patients [15,16,17,18]. Recent research on the dynamics of the autonomic nervous system (ANS) in pathological conditions has demonstrated the existence of an autonomic imbalance in orthostatic syndrome [19], respiratory disease [20], and cardiovascular pathologies [21] by mathematical modeling of physiological interactions related to system coupling and causality. However, scarce studies have addressed the study of ANS dynamics in COVID survivors. In this regard, the pulse–respiration quotient (PRQ) has proven to be a powerful parametrization tool for analyzing the complex behavior of the ANS and its interactions in a wide range of clinical and research settings [22].

Slow-paced breathing has been introduced as a relaxation technique that consists of reducing the breathing rate (BR) to about 6 cycles per minute (0.1 Hz), usually with 5-s deep inhalation and exhalation phases [23]. This breathing technique, also referred to as diaphragmatic [24] or abdominal slow-paced breathing [25], has shown a beneficial impact in both healthy and ill individuals [26,27]. Other breathing exercises, such as meditation and mindfulness, have been shown to reduce depression, pain, anxiety, and stress indicators [28]. Recent reports have suggested that increased respiratory sinus arrhythmia (RSA) mediated by a lower breathing rate improves baroreflex sensitivity and increases parasympathetic nervous control of the heart [29,30]. However, there is limited evidence on the effects of psychophysiological assessments on the ANS response of COVID-19 survivors, and the ongoing studies do not include individuals with concomitant disease [31]. A relevant study highlights the importance of assessing autonomic function to detect possible alterations that may compromise the recovery of COVID-19 survivors and to design biofeedback training based on their characteristics [32].

Thus, this work aimed to study the cardiorespiratory dynamics by linear and nonlinear analysis of photoplethysmographic and respiratory time series on COVID-19 survivors under a psychophysiological assessment (study I). In addition, as a secondary exploratory study, the effect of different comorbidities on the cardiorespiratory dynamics of COVID-19 survivors was examined (study II). We hypothesized that relaxation induces increased cardiorespiratory parasympathetic activity in COVID-19 survivors.

## 2. Materials and Methods

### 2.1. Subjects

This exploratory study involved 49 survivors of COVID-19 illness (32 men and 17 women) aged 27 to 84 years old (49 ± 12 years) who were hospitalized between 2020 and 2022 at the Instituto Nacional de Enfermedades Respiratorias (INER, Mexico City) due to COVID-19 complications in the acute infection phase. All the participants were laboratory-confirmed with SARS-CoV-2 and surveyed at three months post-hospital discharge. Table 1 depicts the demographic and clinical characteristics of the sample; psychological disorders such as stress, anxiety, and depression were the most common among participants (men: 9; women: 11) who disclosed having sequelae after the infection, while most patients reported at least one cardiovascular, pulmonary, motor, cognitive, or other symptoms in the aftermath of the disease.

The research protocol was approved by the Ethics Committee of the National Institute of Respiratory Diseases, Mexico (approval number C57-21). All volunteers in this study signed an Informed Consent form when they agreed to participate, and all methods were performed following the relevant guidelines and regulations. The exclusion criteria involved limited mobility and receiving treatment at home instead of being hospitalized. The sample size for this exploratory research was determined by convenience sampling.

### 2.2. Psychophysiological Assessment (Protocol for Relaxation)

A validated protocol of 10 min for relaxation [32,33] was conducted on COVID-19 survivors in a quiet room in the Coordination of Psychology of the Cardiology Department of the INER between 8 a.m. and 12 p.m. to avoid major fluctuations associated with circadian rhythms [34,35]. The protocol for relaxation involved four phases with the same temporal duration: (1) open eyes; (2) closed eyes; (3) natural relaxation, and (4) induced relaxation. The specific duration and tasks of the participants in each phase are shown in Table 2. In addition, participants were asked to sit comfortably positioned sitting upright in a chair, resting their hands on their legs without taking their feet off the ground or moving while recording was performed.

The COVID-19 survivors were instructed to follow all the phases in this protocol. Before the test, the medical staff guided the patient on how to perform a paced-breathing technique at a fixed rate of six breaths/minute with a 1:1 ratio of inhalation to exhalation time (5 s each). Additionally, for the natural relaxation phase, individuals were asked to relax without an explicit indication of modifying their breathing pattern or frequency as requested for diaphragmatic breathing. In this study, the protocol for relaxation belongs to a preliminary psychophysiological phase of examination of participants who subsequently were enrolled in biofeedback therapy based on their heart rate and breathing rate values.

### 2.3. Data Acquisition and Pre-Processing Process of Photoplethysmographic and Respiratory Time Series in COVID-19 Survivors

Simultaneously to the protocol for relaxation, the COVID-19 survivors were instrumented with a finger-clip pulse oximeter sensor on the thumb of the left hand, followed by two adjustable sensor belts, one on the chest and another on the abdomen (Figure 1). The sensors belong to a ProComp Infiniti System (Thought Technology Ltd., Montreal, CA, Canada) with BVP-Flex/Pro (photoplethysmography sensor/pulse oximeter) and Resp-Flex/Pro (respiration sensor belt for chest and abdominal recordings). The pulse and respiratory signals were recorded with a sampling frequency of 2048 Hz and 256 Hz, respectively. The photoplethysmographic (PPG) and respiratory (RESP) signals were recorded without pauses among phases and had a total duration of 10 min, with 2.5 min for each task (Figure 1 and Figure 2).

The physiological data recordings were exported as text files to Matlab R2022a (The MathWorks Inc., Natick, MA, USA). The extracted raw RESP signals were filtered with a second-order IIR Butterworth bandpass filter with a bandpass frequency range of 0.05 Hz to 1 Hz. Subsequently, a fourth-order IIR Butterworth low-pass filter with a cutoff frequency of 5 Hz was used to filter the raw PPG signals to remove noise interference.

### 2.4. Data Analysis

After the pre-processing stage, the PPG and RESP time series with more than 10% loss of information at any phase were discarded. In addition, RESP signals with an abnormal breathing pattern were not considered for further analyses. In consequence, forty-nine signals were included for breathing rate variability (BRV), pulse rate variability (PRV), and pulse–respiration quotient (PRQ) data analyses, as indicated in Figure 3.

In this study, two main stages were assessed. In study I, forty-nine signals were included for breathing rate variability (BRV), pulse rate variability (PRV), and pulse–respiration quotient (PRQ) data analyses, as indicated in Figure 3. Furthermore, in study II, as a complementary and exploratory study, we separated the individuals by pre-existing comorbidities. To avoid the influence of sex hormones in the autonomic vagal activity predominantly found in young women [22,36], the group was reduced to 21 male individuals sorted by comorbidities into four categories: 1. patients with pulmonary disease (PULM), 2. patients with cardiovascular disease (CARD), 3. patients with diabetes (DIAB), and 4. individuals with no comorbidities (NOC).

#### 2.4.1. Breathing Rate Variability (BRV)

Peak detection was performed to RESP signals from the local inspiration maxima of each phase to extract the breath-to-breath (BB) time series (expressed in seconds) [37]. The mean breathing rate (mean BR), the standard deviation of BB intervals (SDBB), and the root mean square of successive differences of BB intervals (RMSSD) were computed as proposed by Soni and Muniyandi [38].

#### 2.4.2. Pulse Rate Variability (PRV)

A peak detection analysis was performed on PPG signals to obtain the pulse-to-pulse time series (IBI) expressed in milliseconds (ms); it is reported as pulse rate variability (PRV) since the data were obtained from pulse recordings instead of ECG signals [39,40]. The linear and nonlinear indices (see Table 3) were computed using the Kubios HRV Standard 3.5 software (Kubios Oy, Kuopio, Finland) [41].

#### 2.4.3. Pulse–Respiration Quotient (PRQ)

The pulse–respiration quotient (PRQ) is a parameter reintroduced by Scholkmann and Wolf, obtained from the ratio of the heart rate to the breathing rate that effectively reflects the changes in the cardiorespiratory dynamics mediated by age, sex, physical and cognitive activity, body posture, and chronobiological state, and it is sensitive to the activity executed at the time of the recording [22]. It can be computed from the instantaneous heart rate and breathing rate, or the mean value of both parameters as follows:(1)$$\mathrm{PRQ}=\frac{HR}{BR}\;\;\frac{beats/min}{breaths/min}$$

As proposed by Scholkmann and Wolf, all data regarding the PRQ index are reported as daytime functional PRQ (D-f-PRQ) [22] since our recordings were collected in the morning.

Instantaneous heart rate and breathing rate time series were obtained from IBI and BB time series, respectively. To ensure equidistant signals, the instantaneous breathing rate time series (Figure 4b) was upsampled to match the equivalent value of the breathing rate at the same instant of the beat occurrence (Figure 4a), both signals were then used for calculating the instantaneous PRQ. Figure 5 depicts the PRQ time series computation from the instantaneous heart rate (HR) and breathing rate (BR) signals for the four phases of the protocol for relaxation (Table 2).

The PRQ analysis included the mean PRQ and standard deviation of instantaneous PRQ (SD-PRQ) as the linear indexes, followed by sample entropy (SampEn) and fuzzy entropy (FuzzyEn) as the nonlinear parameters; the computing of the nonlinear indexes is described below.

##### Sample Entropy (SampEn)

Sample entropy is a widely known parameter used to evaluate how regular the fluctuations of a time series are, reducing the measured bias usually found when using approximate entropy (ApEn) associated with the signal length and self-matches [43]. This method described by Richman and Moorman is computed as follows:

From a vector $x_{N}=\left\{x_{1},x_{2},x_{3},\ldots ,x_{N}:1\leq i\leq N\right\}$ of N samples, the vectors $u\left(i\right)$ are formed as a series of *m* successive points:(2)$$u\left(i\right)=\left[x\left(i\right),x\left(i+1\right),x\left(i+2\right),\ldots ,x\left(i+m-1\right)\right],$$
*m* being the embedding dimension of the vector length. From that, the distance d, defined as the maximum absolute difference of the elements of the vectors is computed; two points are similar when d between them is less than r:(3)$$d\left[u\left(i\right),u\left(j\right)\right]=\max\left\{\left|u\left(i+k\right)-u\left(j+k\right)\right|:0\leq k\leq m-1\right\}\leq r$$
where r is the tolerance factor and equals to 0.1–0.2 times the standard deviation SD of $x_{N}$ when the data are not normalized [43,44,45,46].

Then, we calculate the number of times the distance of $u\left(j\right)$ from $u\left(i\right)$ vector is less or equal to r, for which j≠i , j≤N−m+1, to avoid self-matches:(4)$$C_{i}^{m}\left(r\right)={\left(N-m+1\right)}^{-1}\overset{N-m}{\underset{i=1}{\sum }}number\;of\;times\;that\;d\left[u\left(i\right),\;u\left(j\right)\right]\leq r$$

The average similarity Φ is then determined as:(5)$$\Phi^{m}\left(r\right)=\frac{1}{N-m+1}\sum_{i=1}^{N-m+1}C_{i}^{m}\left(r\right).$$

Therefore, SampEn is obtained by:(6)$$SampEn\left(m,r,N\right)=-\ln\left(\frac{\Phi^{m}\left(r\right)}{\Phi^{m+1}\left(r\right)}\right)$$

##### Fuzzy Entropy (FuzzyEn)

Fuzzy entropy, introduced by Cheng et al. and based on the fuzzy set theory proposed by De Luca and Termini, evaluates the complexity and ambiguity of a given data set [47,48]. This measure is more reliable for short-length data and is less affected by noise [49]. FuzzyEn is computed as follows:

From a time series of N samples $\left\{u\left(i\right):1\leq i\leq N\right\}$, vectors of m length are formed as follows:(7)$$X_{i}^{m}=\left\{u\left(i\right),u\left(i+1\right),\ldots ,\;u\left(i+m-1\right)\right\}-u_{0}\left(i\right)\;,\;\;i=1\;,\;\ldots \;,\;N-m+1,$$
where $X_{i}^{m}$ is conformed of m successive u values, and $u_{0}\left(i\right)$ represents the subtraction of the baseline of each vector to normalize the data:(8)$$u_{0}\left(i\right)=m^{-1}\overset{m-1}{\underset{j=0}{\sum }}u\left(i+j\right).$$

Afterward, $d_{ij}^{m}$ is defined as the maximum absolute difference between the elements of $X_{i}^{m}$ and $X_{j}^{m}$:(9)$$d_{ij}^{m}=d\left[X_{i}^{m},X_{j}^{m}\right]=\max\left|\left(u\left(i+k\right)-u_{0}\left(i\right)\right)-\left(u\left(j+k\right)-u_{0}\left(j\right)\right)\right|\;:\;\;0\leq k\leq m,$$
from which the similarity degree is obtained by the fuzzy function with given n and r parameters:(10)$$D_{ij}^{m}\left(n,r\right)=\mu \left(d_{ij}^{m},n,r\right)=\exp\left(-\frac{{\left(d_{1j}^{m}\right)}^{n}}{r}\right)\;,\;i\neq j$$

Then, the average similarity expressed by $\varphi^{m}\left(r\right)$ is computed:(11)$$\varphi^{m}\left(r\right)=\frac{1}{N-m}\overset{N-m}{\underset{i=1}{\sum }}\left(\frac{1}{N-m-1}\overset{N-m}{\underset{j=1,\;j\neq i}{\sum }}D_{ij}^{m}\right),$$
to later determine the FuzzyEn as:(12)$$FuzzyEn\left(m,n,r,N\right)=\ln\varphi^{m}\left(r\right)-\ln\varphi^{m+1}\left(r\right)$$

These entropies were evaluated using PyBios 1.0.0 software (Ribeirão Preto, SP, Brazil), setting the following parameters for sample entropy (*m* = 2, *r* = 0.2) and fuzzy entropy (*m* = 2, *r* = 0.2, *n* = 2) [50].

### 2.5. Statistical Analysis

The mean values of linear and nonlinear indices obtained from BB, IBI, and PRQ time series were evaluated by a Lilliefors (Kolmogorov–Smirnov) test to assess normal distribution. Subsequently, multiple comparisons between phases with respect to Phase 4 were performed; paired *t*-tests were conducted if data met the normality assumption; otherwise, we conducted a non-parametric Wilcoxon signed-rank test instead. A non-parametric Kruskal Wallis test was carried out for all indices extracted from BRV, PRV, and PRQ analysis at the induced relaxation phase for comorbidity-based statistical analysis. All these statistical tests were performed using Matlab R2022a software (The MathWorks Inc., Natick, MA, USA).

## 3. Results

### 3.1. Breathing Rate Variability

A lower significant mean value of the respiration rate was found in Phase 4 (9.3 ± 3.6 breaths/min) compared to the other three Phases (*p* < 0.001) following a slow breathing pattern (Figure 6a). It confirmed that individuals reduced the number of breaths per minute. Meanwhile, the SDBB and RMSSD increased significantly (*p* < 0.001) in Phase 4 compared to the other three Phases (Figure 6b,c).

### 3.2. Pulse Rate Variability

#### 3.2.1. Linear Features

A significant increase (*p* < 0.001) is exhibited for SDNN and LF/HF ratio in Phase 4 compared with the other three phases. Figure 7 compares the linear indexes of each phase from the PPG time series.

#### 3.2.2. Nonlinear Features

The nonlinear features exhibited significantly decreased mean values for Phase 4: SampEn and $a_{2}$ long-term fluctuations decrease when breathing at a 0.1 Hz rate (Figure 8a,c) rather than following the other three breathing methods. However, short-term fluctuations ($a_{1}$) seemed to stand out in Phase 4 (Figure 8b) from the rest of the stages.

### 3.3. Pulse–Respiration Quotient (PRQ) Variability

From the instantaneous PRQ time series, Phase 4 reported increased mean PRQ and SD-PRQ values compared to the rest of the stages (*p* < 0.001, Figure 9a,b). Concerning the nonlinear analysis, we observed a significant decrease in both entropy indexes (*p* < 0.001) at Phase 4 of the protocol (Figure 9c,d).

The mean and standard deviation of all evaluated indexes from RESP (Table A1), PPG (Table A2), and PRQ (Table A3) time series are listed in Appendix A to show the main changes among protocol phases. The table also includes the *p*-values of the statistical test when comparing the first three phases with the last phase. All reported parameters showed a statistical significance with *p* < 0.05.

### 3.4. Comorbidity-Based Analysis

We reported the differences among comorbidity groups at the slow breathing phase. The features of these groups are listed in Table 4. We observed a mean heart rate difference with statistical significance (*p* = 0.03) between the no comorbidities (NOC) and diabetic (DIAB) groups (depicted in Figure 10a) but no significant variation with cardiovascular (CARD) and lung disease (PULM) groups.

Although the *p*-value obtained from a Kruskal–Wallis test for the mean PRQ did not show a significant difference, a trend (*p* = 0.0748) was observed among diabetic individuals compared to those with no pre-existing comorbidities (Figure 10b).

## 4. Discussion

In this study, we analyzed whether the diaphragmatic slow-paced breathing technique improves the cardiorespiratory dynamics of COVID-19 survivors. In addition, we evaluated whether the patients’ comorbidities significantly affect pulse and respiration rates.

### 4.1. Breathing Rate Variability

Firstly, BRV was studied to demonstrate that metrics obtained from RESP signals show differences among phases (Figure 6A). For Phases 1 to 3, we found lower values in SDBB and RMSSD-BB (less than 1 s). This finding is consistent with Alhuthail et al. [51], who found RMSSD and SDBB values less than 1 s in post-COVID subjects using structured light plethysmography. Additionally, we confirmed that the psychophysiological assessment modified the mean breathing rate in Phase 4 compared to the other three phases. Moreover, we found that SDBB (Figure 6b) and RMSSD (Figure 6c) indices increased in Phase 4 compared to the previous phases; both indices are indicators of the total variability of breathing and the short-term variability between breathing cycles, respectively [38]. Research on BRV in COVID-19 survivors is scarce, and even less is known about the impact of diaphragmatic breathing in this population. Particularly, relevant research has been published recently about the impact of meditation on breathing rate and BRV [28,52,53]. Some authors have reported increased RMSSD related to breathing awareness in meditator and non-meditator participants [38]. This observation may partially explain our findings.

### 4.2. Pulse Rate Variability

From the linear analysis of PRV, we found that slow-paced breathing (Phase 4) significantly increases cardiac parasympathetic activity and global variability, depicted by SDNN and RMSSD-PP values (Table A2, Appendix A). This finding is consistent with You et al. [54], who found an increase in the RMSSD during slow-paced breathing at six breaths per minute compared to the baseline condition in healthy subjects. In the frequency analysis, we found that slow-paced breathing increased the LF power band and decreased the HF power band in Phase 4 compared to previous phases (Table A2). These findings are consistent with the findings of Li et al. [55] on uncorrected spectral HRV analysis, which showed lower HF power, increased LF band, and LF/HF ratio in patients with hypertension under 8 breaths/min breathing. In contrast, the study published by Sakakibara [25] found increased HRV during paced breathing at eight breaths per minute in healthy individuals with an increased HF power band. A possible explanation for this might be that 43% of our participants were COVID-19 survivors with no comorbidities, while in the study by Li et al., the pattern was shown in the group with hypertension. Hypertension reduces the activity of the parasympathetic branch of the ANS and increases sympathetic activity [56,57]. Moreover, the observed decrease in the HF power band in our group in Phase 4 could be partially explained by the autonomic imbalance reported in COVID-19 survivors.

In the nonlinear analysis, we observed a decrease in SampEn during slow-paced breathing. This parameter measures the regularity and complexity of a time series, with lower values indicating more regular patterns [43]. In our group, this index provides evidence that slow-paced breathing alters the regulation of the cardiac system. We did not recruit a control group comprising healthy subjects to evaluate possible autonomic impairment. However, other authors have reported that COVID-19 survivors exhibit lower SampEn values, which reflect a higher degree of autonomic impairment, compared to healthy control groups [58,59]. In this study, the SampEn values are higher than those reported previously. A possible explanation for this difference might be due to the inclusion of participants with diabetes and cardiovascular diseases (as shown in Table 1 and Table 4). Bajić et al. [59] found that patients with diabetes and hypertension exhibited higher sample entropy among the comorbidities listed.

Previous studies have reported a distinct behavior of the DFA scaling exponents in COVID-19 survivors compared to a healthy group. Our results partially agree with those reported by Kurtoğlu et al. [58]. Our group demonstrated that the long-term scaling exponent α_2_ value was lower than theirs for all phases. This difference can be partially explained by our small sample size, compared to other studies, and the comorbidities present in our group. Additionally, a study concluded that diabetic individuals exhibited an increased α_2_ compared to the short-term exponent α_1_, which indicates decreased parasympathetic activity. Conversely, healthy subjects showed the opposite [60].

Referring to Phase 4, we observed that slow-paced breathing modifies the DFA exponents, with an increase in the short-term exponent α_1_ and a decrease in the long-term exponent α_2_ (Figure 8b,c), which may be related to the reciprocal regulation between the cardiac and respiratory systems, influenced by slow breathing or an individual’s posture, which enhances the parasympathetic influence on cardiac function. Matić et al. found that short-term scaling α_1_ increased significantly, while the long-term scaling exponent α_2_ decreased in standing and during slow breathing, indicating reciprocal regulation. Their study showed that changes in posture and breathing patterns significantly impact cardiac and respiratory interactions mediated through sympathovagal control [61]. Our results demonstrated that nonlinear indices (SampEn and DFA) had lower *p*-values than linear indices (Table A2, Appendix A).

### 4.3. Pulse–Respiration Quotient

The PRQ analysis reveals complex cardiorespiratory interactions that are not perceptible through the single analysis of BRV and PRV [22]. Currently, studies related to PRQ primarily involve healthy individuals, and there is no data regarding COVID-19 survivors who experienced post-acute syndrome or performed slow-paced breathing. Matić et al. reported increased mean PRQ and PRQ variability with orthostasis and slow-paced breathing as signs of cardiorespiratory coupling regulation and adaptability [62]. Our findings on the mean PRQ (Figure 9a) and SD-PRQ (Figure 9b) showed a significant difference in response to the change in breathing frequency with higher statistical significance than the mean and SD of PRV indices. However, contrary to this study, our mean PRQ and SD PRQ indices exhibited a slightly lower *p*-value than SD RESP (Table A3).

We found that lower values of SampEn (Figure 9c) and FuzzyEn (Figure 9d) of the PRQ time series might indicate higher cardiorespiratory coupling mediated by the ANS in the induced-relaxation phase, as suggested by Tian and Song, who reported a decrease in the FuzzyEn index and BR during slow breathing in healthy participants [63].

### 4.4. Comorbidity-Based Analysis

Related to the effects of slow breathing on COVID-19 survivors with a previously detected comorbidity, we found that even though changes that enhance autonomic regulation were visible when performing diaphragmatic breathing, patients with type 2 diabetes mellitus (T2DM) still showed a higher heart rate (Figure 10a), possibly associated with cardiac autonomic neuropathy [64,65]. Benichou et al. reported that T2DM patients showed a decreased value of all HRV parameters, which exhibited restricted sympathetic and parasympathetic activity [66]. Another study demonstrated a short-term reduction of stress levels, enhanced glycemic status (reduced hyperglycemia), and increased immune response in diabetic patients with COVID-19 who followed a relaxation technique [67].

Vanzella et al. [68] assessed the changes in HRV among healthy individuals and patients with COPD; a significant decrease in time (RMSSD, SDNN) and frequency (LF, HF) domain indices of patients with COPD were found, as indicators of the reduction of both sympathetic and parasympathetic activity that suggests a worsened autonomic modulation. The revision conducted by Hamasaki concluded that diaphragmatic breathing improved the respiratory function of patients with COPD, reduced stress and anxiety symptoms by stimulating vagal activation, and enhanced cardiorespiratory performance in patients with heart failure [69].

Lachowska et al. reported that HR and blood pressure responses decreased considerably with slow breathing exercises in heart failure patients with reduced ejection fraction and enhanced a variety of health-related scores of quality of life (QoL) [70], while another study exhibited improvement in the long-term cardiorespiratory capacity of these patients [71]. Post-stroke patients also seem to benefit from slow breathing, with increased baroreflex sensitivity, lower systolic blood pressure, and HR with increased HRV [72].

Apart from the diabetic group, no significant difference was found among comorbidities and healthy individuals during slow breathing (Figure 10a,b). We supposed that these results could be associated with the fact that both conditions (comorbidities and post-COVID-19 syndrome) have been proven to modify autonomic function, and the reduced sample size may cause no statistically significant difference among comorbidity groups.

### 4.5. Limitations and Future Work

In this study, we encountered some limitations; first, the reduced sample size to address autonomic changes among patients with comorbidities and healthy individuals, where both groups coursed COVID-19 sequelae, and the lack of previous research on the use of the PRQ to evaluate cardiac and respiratory interactions in a similar population to the one presented in this work, that must be considered in the interpretation of the provided data. Despite participants reducing their BR, we also noted that several did not meet the 6 breaths/min breathing rate (Figure 6a and Table A1) that can be partly associated with the group comorbidities and symptoms of post-COVID-19 syndrome. Another possible explanation is that many participants did not have previous training in diaphragmatic breathing.

COVID-19 sequelae negatively impact survivors’ physical and mental health in the short and long term [73,74,75]. It has been observed that most of these patients exhibited high levels of cellular inflammation and autonomic dysregulation [76]. Slow breathing practice has been proven to enhance the autonomic regulation of individuals with diverse chronic illnesses [77]; it improves cardiovascular response, decreases sympathetic hyperactivation, and induces a state of physiological balance, which may be a critical factor in the prevention or treatment of the cardiovascular manifestations [78,79,80]. Additionally, it reduces anxious and depressive symptoms, which can worsen the state of health and recovery of these patients [81,82]. This psychophysiological evaluation allows observing the response to various conditions to identify adaptive and maladaptive response patterns to intervene and, if necessary, modify them [31,81].

Additional studies may be conducted to assess the long-term effect of slow breathing and other psychophysiological interventions on COVID-19 survivors who present post-infection sequelae and may or may not have other comorbidities. Moreover, further analysis of the differences between men and women is suggested with an increased sample size. However, the study of the PRQ index seems like a feasible technique to evaluate cardiac and respiratory interactions and the changes mediated by breathing rate modifications.

## 5. Conclusions

Our results demonstrate that implementing a psychophysiological assessment based on slow paced-breathing induces significant changes in the breathing rate variability and pulse rate variability of COVID-19 survivors. These changes may improve the cardiorespiratory dynamics of this population by enhancing cardiorespiratory coupling in the short-term via increased vagal activity. Additionally, our findings suggest that patients with comorbidities, such as type 2 diabetes mellitus, may still benefit from diaphragmatic breathing despite changes in sympathovagal balance. However, further research is needed to confirm the efficacy of this practice in the long term.

## Figures and Tables

**Figure 1 entropy-25-00874-f001:**
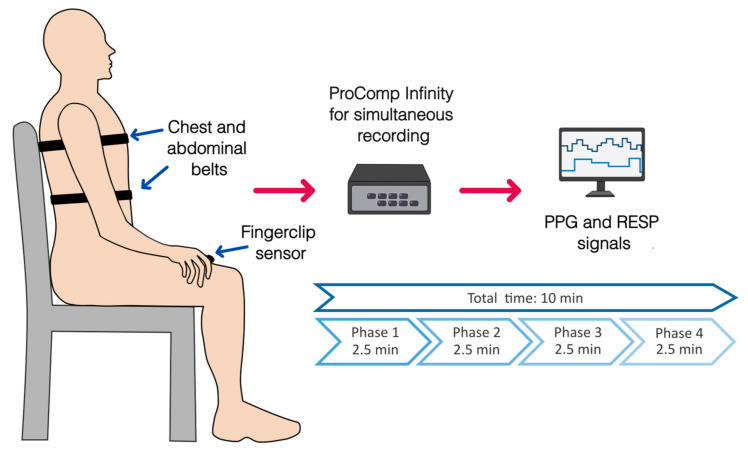
Placement of sensors for physiological recordings of photoplethysmography (PPG, finger clip pulse oximeter) and respiratory signals (RESP, chest, and abdominal belts) during the complete protocol for relaxation. An overview of the phases of the protocol and its duration is shown.

**Figure 2 entropy-25-00874-f002:**
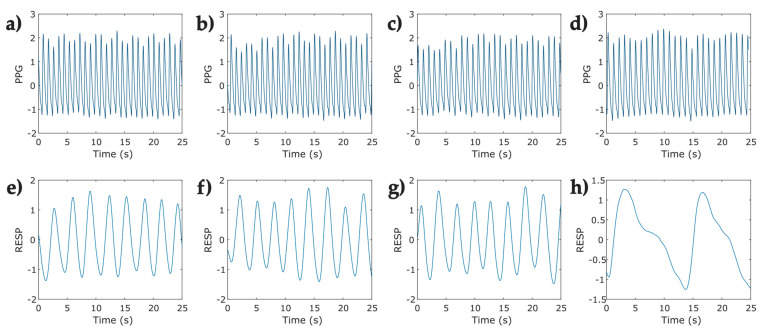
Representative segments (25 s) of photoplethysmographic (PPG) and respiratory signals (RESP) corresponding to one participant in the protocol for relaxation applied to COVID-19 survivors. (**a**) open eyes Phase—PPG signal; (**b**) closed eyes—PPG signal; (**c**) natural relaxation—PPG signal; (**d**) induced relaxation—PPG signal; (**e**) open eyes—RESP signal; (**f**) closed eyes—RESP signal; (**g**) natural relaxation—RESP signal; and (**h**) induced relaxation—RESP signal.

**Figure 3 entropy-25-00874-f003:**
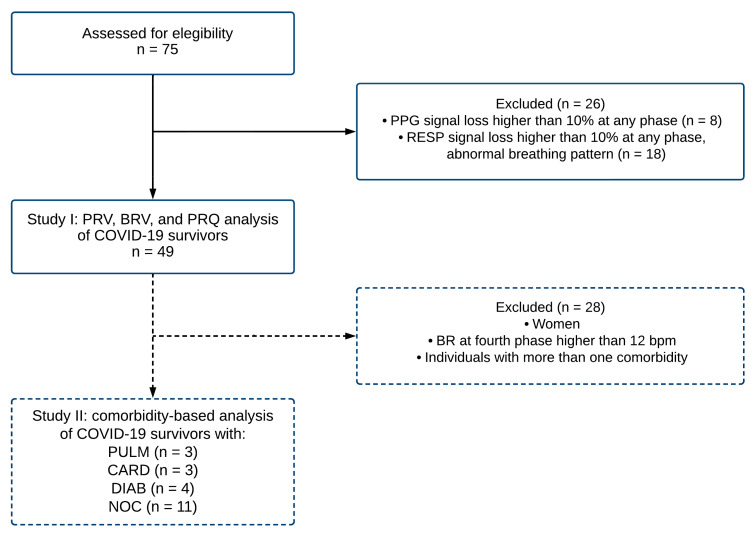
The flowchart describes the procedure for sample selection criteria for the photoplethysmographic (PPG) and respiratory (RESP) signals corresponding to COVID-19 survivors. A total of forty-nine records were included (study I) in the analysis of pulse rate variability (PRV), breathing rate variability (BRV), and pulse–respiration quotient (PRQ). Subsequently, a subsample of twenty-one individuals was evaluated for a comorbidity-based analysis (study II). Note: BR breathing rate; bpm beats per minute; PULM pulmonary disease; CARD cardiovascular disease; DIAB diabetes; NOC no comorbidities.

**Figure 4 entropy-25-00874-f004:**
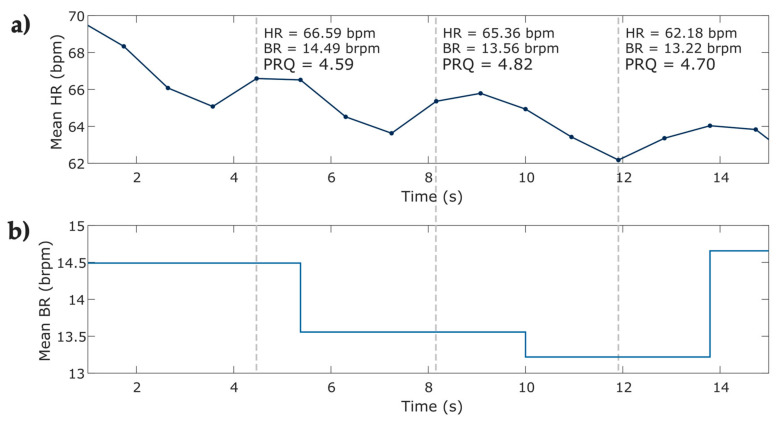
Visual representation of the computing of instantaneous PRQ index from the instantaneous heart and breathing rate from one subject at phase 1 of the protocol. (**a**) 15 s of the instantaneous heart rate time series; (**b**) 15 s of the upsampled instantaneous breathing rate time series. The figure shows HR heart rate; bpm beats per minute; BR breathing rate; brpm breaths per minute; pulse–respiration quotient (PRQ).

**Figure 5 entropy-25-00874-f005:**
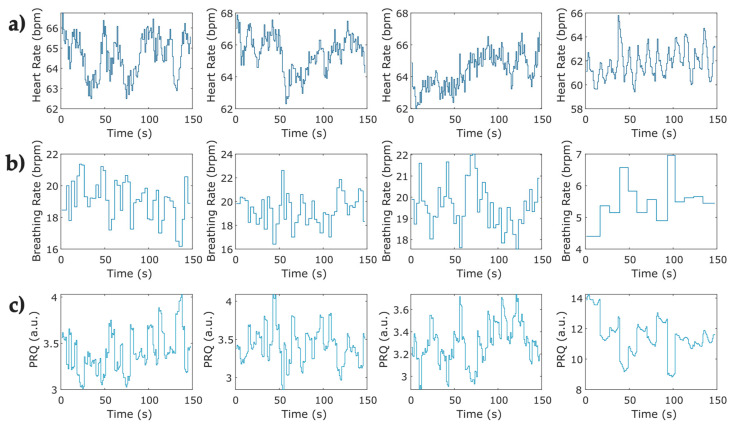
Representative examples of pulse–respiration quotient time series computed from instantaneous heart rate and breathing rate for the four phases of the protocol for relaxation in one COVID-19 survivor. (**a**) Instantaneous heart rate at the four phases obtained from a sample of photoplethysmographic record; (**b**) sample of the instantaneous breathing rate at four phases of the protocol obtained from breathing signals; (**c**) instantaneous pulse–respiration quotient computed from instantaneous HR and BR series (**a**,**b**).

**Figure 6 entropy-25-00874-f006:**
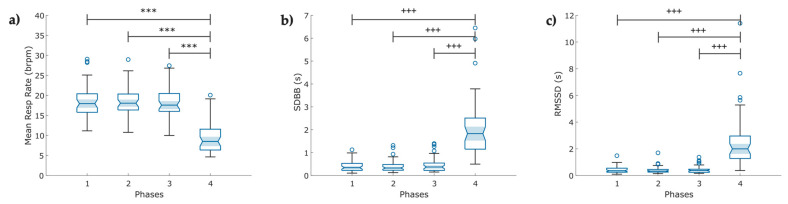
Boxplot of indices of the breathing rate variability (BRV) extracted from the respiration signal (RESP) in COVID-19 survivors (N = 49) under a psychophysiological assessment of four phases: (**a**) mean respiration rate; (**b**) standard deviation of breath-to-breath (BB) intervals (SDBB); and (**c**) root mean square of the successive difference between breath intervals (RMSSD). The figure shows: bpm breaths per minute, ৹ outliers, *** *p* < 0.001 obtained by paired *t*-test, **^+++^**
*p* < 0.001 obtained by Wilcoxon signed-rank test.

**Figure 7 entropy-25-00874-f007:**
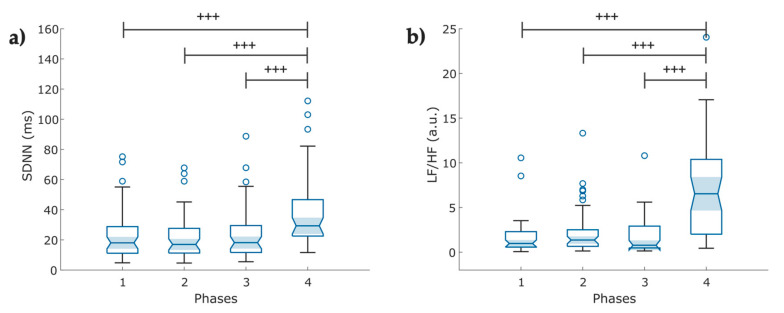
Boxplot of linear indices of the Pulse Rate variability (PRV) extracted from the photoplethysmographic signals (PPG) in COVID-19 survivors (N = 49) under a psychophysiological assessment of four phases: (**a**) mean values of the standard deviation of the IBI of typical sinus beats (SDNN); (**b**) ratio of low frequency (LF) and high frequency (HF) power. The figure shows ৹ outliers, **^+++^** *p* < 0.001, obtained by Wilcoxon signed-rank test.

**Figure 8 entropy-25-00874-f008:**
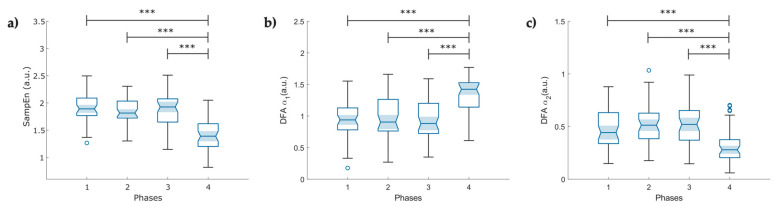
Boxplot of nonlinear indices of the pulse rate variability (PRV) extracted from the photoplethysmographic signals (PPG) in COVID-19 survivors (N = 49) under a psychophysiological assessment of four phases: (**a**) sample entropy (SampEn); (**b**,**c**) detrended fluctuation analysis between successive PP intervals for short ($a_{1}$) and long-term ($a_{2}$) fluctuations, respectively. The figure shows ৹ outliers, *** *p* < 0.001, obtained by paired *t*-test.

**Figure 9 entropy-25-00874-f009:**
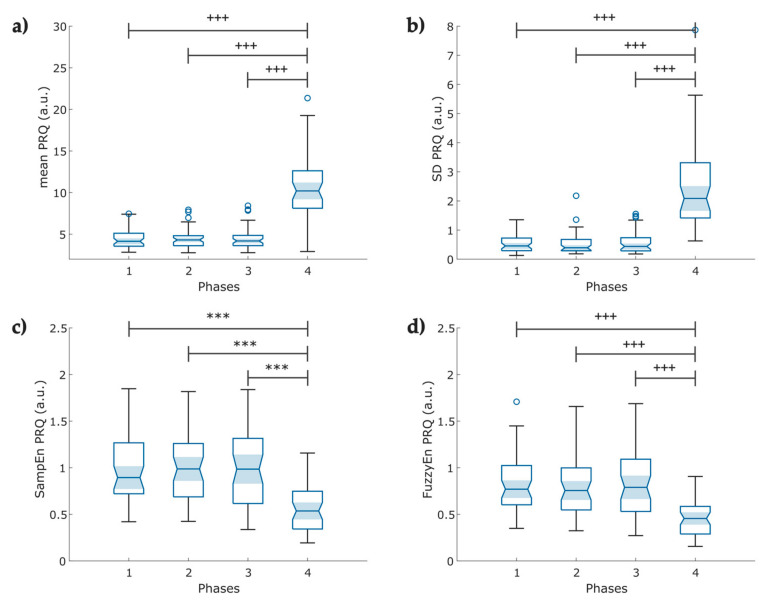
Boxplot of pulse–respiration quotient (PRQ) indexes obtained from the instantaneous heart rate and breathing rate time series (N = 49); (**a**) Mean PRQ, during slow breathing (6 breaths/min), a significant increase in PRQ is observed; (**b**) fluctuations of the instantaneous PRQ reported as the PRQ standard deviation; (**c**) Sample entropy of the instantaneous PRQ; (**d**) Fuzzy entropy of the PRQ. The figure shows ৹ outliers, *** *p* < 0.001 obtained by paired *t*-test, **^+++^** *p* < 0.001 obtained by Wilcoxon signed-rank test.

**Figure 10 entropy-25-00874-f010:**
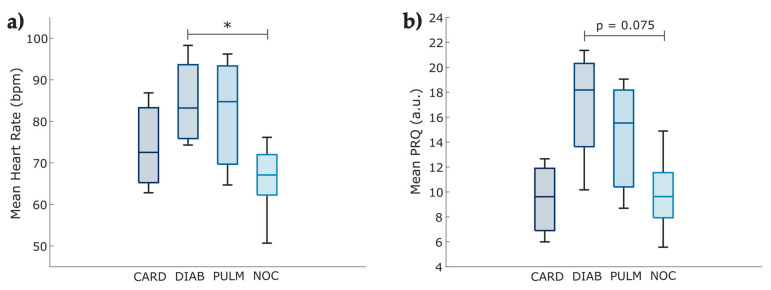
Boxplot of the parameters evaluated by individuals’ comorbidities, comparing the Phase 4 obtained from the photoplethysmographic and pulse–respiration quotient time series of COVID-19 survivors (N = 21). (**a**) The mean heart rate changes among the comorbidity groups in the fourth phase of the test. (**b**) The mean PRQ index comparison among the cardiovascular disease (CARD), diabetes (DIAB), lung disease (PULM), and no comorbidities (NOC) groups in the fourth phase of the test. The figure displays bpm (beats per minute), with * indicating *p* < 0.05 obtained by the Kruskal–Wallis test.

**Table 1 entropy-25-00874-t001:** Demographic and clinical characteristics of the patients.

Variable	Sample (n = 49)	Men (n = 32)	Women (n = 17)
No comorbidities	26 (53%)	15 (47%)	11 (65%)
≥1 comorbidity	23 (47%)	17 (53%)	6 (35%)
Mean hospital stay	21 days	22.4 days	19.5 days
No comorbidities	15.6 days	14.8 days	16.6 days
≥1 comorbidity	28 days	29 days	24.8 days
Sequelae	26 (53%)	14 (44%)	12 (71%)
Cardiovascular	2	1	1
Pulmonary	2	1	1
Motor	4	2	2
Psychological	20	11	9
Cognitive	4	2	2
Others	7	2	5

**Table 2 entropy-25-00874-t002:** Protocol for relaxation applied to COVID-19 survivors.

Phase		Duration	Tasks
**1**	**Open eyes**	2.5 min	Sitting upright with open eyes, spontaneous breathing
**2**	**Closed eyes**	2.5 min	Sitting upright with closed eyes, spontaneous breathing
**3**	**Natural relaxation**	2.5 min	Sitting upright with closed eyes, natural skill of relaxation
**4**	**Induced relaxation**	2.5 min	Sitting upright with closed eyes, slow-paced breathing (six breaths/min, 1:1 inhalation to exhalation ratio)

**Table 3 entropy-25-00874-t003:** Parameters calculated from beat-to-beat interval (IBI) times series of COVID-19 survivors performing a protocol for relaxation.

Type	Index
Linear	Mean PP distance
	SDNN
	RMSSD
	LF power
	HF power
	LF/HF ratio
Nonlinear	SampEn
	DFA $\alpha_{1}$
	DFA $\alpha_{2}$

PP—pulse-to-pulse; SDNN—standard deviation of normal pulse intervals; RMSSD—root mean square of successive PP interval differences; LF—relative power of the low-frequency band; HF—relative power of the high-frequency band; LF/HF—ratio of low to high-frequency band power; SampEn—sample entropy; DFA $\alpha_{1}$—detrended fluctuation analysis (short-term); DFA $\alpha_{2}$—detrended fluctuation analysis (long-term). Adapted from [42].

**Table 4 entropy-25-00874-t004:** Clinical characteristics by comorbidity.

Comorbidity	n	Age (Years ± SD)
Cardiovascular disease	3	49 ± 14
Lung disease	3	53.75 ± 13.1
Diabetes	4	50.8 ± 13.8
No comorbidities	11	47.5 ± 13.1

Data presented in this table correspond to the group with n = 21.

## Data Availability

The access to the data presented in this study should be requested of the authors and may be restricted due to hospital confidentiality policy.

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
