# Peer review of "Induced Relaxation Enhances the Cardiorespiratory Dynamics in COVID-19 Survivors"

_entropy, 2023, doi:10.3390/e25060874_

Round 1

Reviewer 1 Report

In this study, the authors explored the cardiorespiratory dynamics in COVID-19 survivors by linear and nonlinear analysis. They reported the findings in BRV, PRV, and PRQ, as well as the SampEn and FuzzyEn. The overall aim and outline of the manuscript are very clear. I have few concerns.

Major:

1. Please make sure the extract number of subjects used in the final study, since the number is not consistent between the Abstract and the main text.

2. It is unclear about the descriptions of data analysis. How to handle the signals? Are they segmented or not? What is the length of each signal?

3. There is no control group included in this study, which may blur the observed results. How about the variation trend or pattern should be in the healthy controls?

Minor:

4. How to understand the meaning of the sentence in line 95 in sec. 2.1?

5. The underline of No. 1 in Table 1 should be omitted.

6. The description of “four groups” in Line 150 should be modified to be more clear. And what are the four groups?

Author Response

Thank you for your review and comments on our paper, please see the attachment.

Reviewer 2 Report

- There is a minor typo error at line 69: "arrhythmia";

- The sample size is really very small, I suggest to increase it;

- Figure 3 is not clear or it has mistake that must be corrected;

- at line 108 the ratio 5:5 should be: 1:1 ratio of inhalation to exhalation repeated 5 times;

- In Table 1 the number "1" should not be underlined;

- at line 149/150 there is written:"...to avoid the influence of biological sex [22,36], the group was reduced to twenty-one male individuals sorted out into four groups." Potentially gender bias - usually to avoid this type of influence is a common practice to include an equal number of both sex or to check if there is a significant difference between the two. Another common practice is to check if there are significant differences between men+women/men/women;

- at lines 149 "the group was reduced to twenty-one male", but in figures 6, 7 and 8 N=49;

-Finally a personal consideration: If the authors would like to test the differences between the 4 breath techniques, they could take in consideration to not perform them consequentially. Otherwise the second breath technique could be influenced by the first and so on

Fine

Author Response

(The authors gave the same response as above.)

Reviewer 3 Report

Dear Ladies and Gentlemen, dear authors, dear editors,

Thank you very much for this interesting work in a relevant field of biopsychosocial medicine. The work chooses a non-drug intervention as a therapeutic step and can show relevant effects on the cardiovascular system and the autonomic nervous system.

However, some information is omitted, which in my opinion should be urgently added before publication. A clearly structured patient characteristic with comorbidities, age and severity of COVID-19 infection should be added. Likewise, a clear representation of the persistent complaints in a table should be attempted to be recorded. This would be another aspect that should be taken into account statistically. A further statistical evaluation with regard to the distance corrected according to the symptoms would be desirable and would provide additional information. Furthermore, Figure 3 remains unclear with regard to inclusions and exclusions. The first 21 in particular plays a decisive role in this. 

Minimal aspects of English should be taken into account, otherwise the work is easy to read.

Author Response

(The authors gave the same response as above.)

Round 2

Reviewer 3 Report

Dear Editor, Dear Authors,

Thank you very much for changes and the commitment so far. From my point of view, the most important points have been taken up and taken into account in the current version of the publication, so that I would recommend that the publication be accepted.